# Film-Forming Systems for the Delivery of DNDI-0690 to Treat Cutaneous Leishmaniasis

**DOI:** 10.3390/pharmaceutics13040516

**Published:** 2021-04-08

**Authors:** Katrien Van Bocxlaer, Kerri-Nicola McArthur, Andy Harris, Mo Alavijeh, Stéphanie Braillard, Charles E. Mowbray, Simon L. Croft

**Affiliations:** 1Department of Biology, York Biomedical Research Institute, University of York, York YO10 5DD, UK; 2Pharmidex Pharmaceutical Services Ltd., London EC2V 8AU, UK; K.McArthur@pharmidex.com (K.-N.M.); A.Harris@pharmidex.com (A.H.); Mo.Alavijeh@pharmidex.com (M.A.); 3Drugs for Neglected Diseases *initiative* (DND*i*), 1202 Geneva, Switzerland; sbraillard@dndi.org (S.B.); cmowbray@dndi.org (C.E.M.); 4Faculty of Infectious and Tropical Diseases, London School of Hygiene & Tropical Medicine, London WC1E 7HT, UK; simon.croft@lshtm.ac.uk

**Keywords:** cutaneous leishmaniasis, topical treatment, film-forming system, DNDI-0690, antileishmanial activity

## Abstract

In cutaneous leishmaniasis (CL), parasites reside in the dermis, creating an opportunity for local drug administration potentially reducing adverse effects and improving treatment adherence compared to current therapies. Polymeric film-forming systems (FFSs) are directly applied to the skin and form a thin film as the solvent evaporates. In contrast to conventional topical dosage forms, FFSs strongly adhere to the skin, favouring sustained drug delivery to the affected site, reducing the need for frequent applications, and enhancing patient compliance. This study reports the first investigation of the use of film-forming systems for the delivery of DNDI-0690, a nitroimidazole compound with potent activity against CL-causing *Leishmania* species. A total of seven polymers with or without plasticiser were evaluated for drying time, stickiness, film-flexibility, and cosmetic attributes; three FFSs yielded a positive evaluation for all test parameters. The impact of each of these FFSs on the permeation of the model skin permeant hydrocortisone (hydrocortisone, 1% (*w/v*) across the Strat-M membrane was evaluated, and the formulations resulting in the highest and lowest permeation flux (Klucel LF with triethyl citrate and Eudragit RS with dibutyl sebacate, respectively) were selected as the FFS vehicle for DNDI-0690. The release and skin distribution of the drug upon application to *Leishmania*-infected and uninfected BALB/c mouse skin were examined using Franz diffusion cells followed by an evaluation of the efficacy of both DNDI-0690 FFSs (1% (*w/v*)) in an experimental CL model. Whereas the Eudragit film resulted in a higher permeation of DNDI-0690, the Klucel film was able to deposit four times more drug into the skin, where the parasite resides. Of the FFSs formulations, only the Eudragit system resulted in a reduced parasite load, but not reduced lesion size, when compared to the vehicle only control. Whereas drug delivery into the skin was successfully modulated using different FFS systems, the FFS systems selected were not effective for the topical application of DNDI-0690. The convenience and aesthetic of FFS systems alongside their ability to modulate drug delivery to and into the skin merit further investigation using other promising antileishmanial drugs.

## 1. Introduction

The leishmaniases are a group of neglected tropical diseases that typically affect populations in low- and middle-income countries. The most prevalent form of this disease is cutaneous leishmaniasis (CL) which is characterised by skin lesions ranging from closed nodules to plaques and ulcers—symptoms typically situated on exposed body parts [1,2]. Current treatments are unsatisfactory and limited due to adverse effects, variable efficacy, cost, logistical challenges [3], and lengthy treatment durations, which sometimes involve invasive drug administration [4]. In contrast to visceral leishmaniasis (VL), where the parasite causes a life-threatening pathology in the liver and spleen, the visibility of the clinical symptoms of CL can lead to patients being ostracised, causing psychosocial stress and a reduced quality of life [5,6]. The inherently different disease severity and biophase (dermal layer of the skin for CL vs. internal organs for VL) of both diseases requires different approaches to drug discovery and development, which is reflected in their diverse target product profiles [7,8]. In localised CL, the *Leishmania* parasites are primarily located in the dermis, favouring local drug administration in order to achieve high skin concentrations with minimal systemic exposure and associated potential adverse effects. High local drug concentrations can be challenging to achieve due to the efficient barrier properties of the stratum corneum. It is especially difficult for conventional topical dosage forms such as creams and gels as they are not designed for sustained drug delivery and often fail to ensure persistent contact with the skin [9].

To date, only one conventional topical treatment (Leshcutan^®^, Teva Pharmaceuticals, Petah Tikva, Israel) is available in Israel to treat CL. It contains the anti-leishmanial paromomycin and a pore-forming permeation enhancing agent, methylbenzethonium chloride [10]. To further increase the contact time between the drug and the parasite, the wound bed of the lesion is debrided prior to ointment application and occlusion—even then, variable efficacy rates have been reported [10,11,12]. Many other more tolerable topical formulations have been developed, but only one (WR 279,396–15% paromomycin–0.5% gentamicin in a hydrophilic vehicle) has reached phase 3 clinical trials [13,14].

As an alternative approach, we have investigated polymeric film-forming systems (FFSs) that are directly applied to the skin and form a thin, cosmetically acceptable, and transparent film as the solvent evaporates. To be effective the active compound must (i) partition from the film into the skin surface, (ii) permeate into and through the different skin layers, (iii) distribute to dermis, and (iv) accumulate into the phagolysosome compartment of macrophages where the intracellular *Leishmania* amastigotes are located. These different processes are influenced by the microenvironment of the skin, the physicochemical properties of the active compound, the properties of the vehicle formulation, and the frequency and methodology of application. 

Film-forming formulations can lead to sustained drug delivery via two mechanisms. In the first, the drug maintains some solubility in the remaining film, which facilitates a prolonged contact time with the skin from a so-called on-skin reservoir [15]. In the second, modification of the formulation upon application to the skin enhances a rapid transfer of the drug into the stratum corneum that subsequently forms an in-skin reservoir [16], from which the drug slowly diffuses to the deeper skin layers. Both strategies facilitate drug delivery to the local skin site over an extended period of time permitting less frequent applications and improving patient compliance [17,18]. The successful application of FFS is exemplified in Lamisil Once (Novartis Consumer Health). This cutaneous solution contains the poorly water-soluble antifungal drug terbinafine (1% (*w/w*)) incorporated in an FFS for the targeted treatment of dermatophytoses in the stratum corneum which leads to an increased contact time between the drug and the skin [19,20]. 

The nitroimidazole DNDI-0690 (Mw 370 g/mol, solubility (phosphate-buffered saline, pH 7.4) ≈ 1.5 ug/mL, log D _(pH 7.4)_ = 2.45) is a drug candidate currently undergoing clinical phase 1 trials for VL that showed similar in vitro potency against a panel of species that cause CL [21,22]. Franz diffusion studies utilising ex vivo mouse skin confirmed the lack of DNDI-0690 (log D _(pH 7.4)_ = 2.45) permeation through both *Leishmania*-infected and uninfected skin when applied as a saturated solution (thermodynamic activity = 1) [23]. These findings confirmed the inability of this topical formulation (50 μL of 0.063% (*w*/*v*) DNDI-0690 in propylene glycol–ethanol (PG–EtOH) (1:1) to reduce the nodule size and skin parasite load in experimental CL when applied twice a day. In contrast, a dose of 50 mg/kg administered orally once daily showed 95% efficacy or more (lesion size [21] and parasite load measured by bioluminescent radiance or qPCR [23]) in the *Leishmania* major–BALB/c model. Skin microdialysis was utilised to further investigate the skin distribution of free unbound DNDI-0690 into the dermis and revealed dermal concentrations of approximately 250 nM two hours after oral administration of a single 50 mg/kg dose [23], demonstrating its consistent and potent activity against both Old World and New World CL models caused by *L. major* and *L. mexicana,* respectively. In this study, we explored the use of FFS as a skin delivery system for the poorly water-soluble DNDI-0690 to treat single and closed CL nodules in BALB/c mice. 

## 2. Materials and Methods 

### 2.1. Ethics

Ethical approval for this human study was obtained from the University of York Dept. of Biology Ethics Committee (Ref KVB201912). Informed consent was obtained from all 8 participants (4 females and 4 males) prior to commencing the study. The animal work was carried out under a U.K. Home Office project licence according to the Animal (Scientific Procedures) Act 1986 and the new European Directive 2010/63/EU. The project licence (PPL P00B3B595) was reviewed by the University of York Animal Welfare and Ethical Review Board prior to submission and consequent approval by the U.K. Home Office.

### 2.2. Materials

The polymers Kollidon 90F (polyvinylpyrrolidone), Kollidon VA64 (vinylpyrrolidone-vinyl acetate copolymers), and Kollicoat SR 30D (polyvinyl acetate dispersion (aq)) were kindly provided by BASF (Ludwigshafen, Germany); Pemulen TR-2-NF (acrylates/C10–30 alkyl acrylate cross polymers) by Lubrizol (Leeds, UK); Eudragit RS PO (ammonio methacrylate copolymer type B) and Eudragit NM30 D (poly(ethyl acrylate-*co*-methyl methacrylate) (2:1)) by Evonik Industries (Darmstadt, Germany); and Klucel LF Pharm (hydroxypropyl cellulose) from Ashland (Schaffhausen, Switzerland). Four plasticisers with variable lipophilicity triethyl citrate (TEC) (log P = 0.12), tributyl citrate (TBC) (log P = 2.83), dibutyl sebacate (DBS) (log P = 4.87), and propylene glycol (PG) (log P = −0.47); solvents (ethanol, acetonitrile); PBS tablets; a model skin permeant hydrocortisone; the polyethersulfone Strat-M synthetic membrane (25 mm diameter); and additional chemicals such as ß-cyclodextrin were purchased from Merck Life Sciences (previously Sigma Aldrich, Dorset, UK). 

### 2.3. Preparation of the Polymeric Solutions

Placebo FFS were prepared by adding the polymer powder with or without 20% (*w*/*w*—relative to the dry weight of the polymer) plasticiser to ethanol (water was included for Eudragit-based films) (Table 1). Polymer concentrations were selected on the basis of manufacturer guidelines and previous studies [9,24]. A magnetic stirrer was introduced and the whole was left to stir overnight until a homogeneous mixture was obtained.

### 2.4. Evaluation of the Formulations

A series of polymers and plasticisers, commonly used in film-forming products, were identified and combined to form placebo FFS formulations that were evaluated visually according to the ratings described below.

Placebo FFS vehicle (before solvent evaporation). Upon visual inspection, the FFS vehicle was rated as clear or opaque, indicating complete or partial polymer dissolution, respectively. The viscosity, an important factor for spray formulations affecting dosing accuracy and ability of the formulation to spread on the skin, was compared visibly to the viscosity of water (low—score 1), glycerol (medium—score 2), and the benchmark FFS Lamisil Once (high—score 3).

FFS film (after solvent evaporation). Parameters related to patient compliance and user-friendliness such as drying time, outward stickiness, film flexibility, and ease of removal of the film with lukewarm water after application of placebo FFS solution to the forearm of a volunteer. The time required for formation of the film (also drying time) was measured as the interval between application of the FFS solution and evaporation of the solvent, assessed by the absence of humidity when applying a cover slide to the film. Film stickiness was evaluated 5 min after the film had formed by applying cotton wool at a standardised and uniform pressure (225 gr./sq. cm, D-squame pressure instrument, Clinical and Derm, Dallas, TX, USA). A low, medium, or high score was attributed where little to no wool, a thin layer, or a dense layer of wool remained, respectively. The flexibility of the film is important for skin fixation, and adherence and was tested by visual inspection of the film for cracks after having stretched the skin in 2 to 3 directions (flexible for no cracking versus non-flexible for cracking or film detaching). Finally, the ease of film removal was tested by wiping down the film 3 times with a cotton pad wetted with lukewarm water. If the film was not removed at all, the removal was classed as difficult, whereas complete or partial removal of the film was rated as easy.

### 2.5. Parasites and Mice

*L. major* parasites (MHOM/SA85/JISH118) were maintained in Schneider’s insect medium (Sigma, Dorset, UK) with 10% (*v*/*v*) heat-inactivated foetal calf serum (HiFCS; Harlan, UK) at 26 °C and passaged weekly by diluting an aliquot of the existing culture into a 25 mL culture flask with fresh culture medium (1:10 dilution). On the day of infection, stationary phase parasite cultures were centrifuged for 10 min at 2600× *g* at 4 °C, counted using a Neubauer hemocytometer, and resuspended in Schneider’s insect medium at a density of 2 × 10^8^ per mL. Each mouse received 200 μL of this suspension by subcutaneous injection in the rump above the tail. Approximately 7 days post-infection, a nodule became visible which was measured daily in 2 perpendicular directions using digital callipers. When the average nodule size reached 5 mm, the mice were either sacrificed for in vitro permeation studies or regrouped for in vivo formulation evaluation.

### 2.6. Permeation Evaluation Using Model Permeant Hydrocortisone

In a first stage, the permeation of hydrocortisone (log P 1.61) from the different FFSs across the Strat-M membrane was assessed. This synthetic membrane (approximately 300 μm thick) is built of a tight top layer mimicking the stratum corneum covering 2 polyether sulfone layers serving as the dermal skin layer and finally a single layer of polyolefin non-woven fabric support as subcutaneous tissue [25]. The FFS vehicle solution was prepared as mentioned above followed by the application of a 1% (10 mg/mL, *w*/*v*) hydrocortisone formulation. Two control vehicles were included: (i) propylene glycol (PG)/ethanol (EtOH) (1:1), as this was the topical vehicle utilised in previous DNDI-0690 studies, and (ii) 100% EtOH, as a control for the solvent of these FFS formulations. A small amount of vacuum grease was applied to the rim of a Franz diffusion receptor and donor compartment in order to seal the Strat-M membrane in addition to the application of a clamp. PBS with 2% ß-cyclodextrin was left to sonicate for 30 min and then introduced into the receptor compartment. A magnet was introduced via the sampling arm, and the Franz diffusion cells were placed on a magnetic stirrer plate in a warm water bath ensuring a constant membrane temperature of 34 °C (±0.5). Using a positive displacement pipette, we applied 100 μL of the drug-spiked FFS, and every hour for the following 8 h, an aliquot (100 μL) of the receptor fluid was removed and analysed. 

The concentration of hydrocortisone in the samples was quantified by reversed-phase HPLC (Shimadzu Prominence with UV detector) using a Luna C18 column (250 × 2.6 mm, 5 μm, Phenomenex, United Kingdom) with guard column. An isocratic mobile phase comprising 60% A (HPLC-grade water spiked with 0.1% (*v*/*v*) formic acid) and 40% B (HPLC-grade acetonitrile) was used. A detection wavelength of 254 nm, a column temperature of 40 °C, and an injection volume of 20 μL were applied. At a mobile phase flow rate of 1 mL/min, hydrocortisone was eluted with a retention time of 5.4 min. Standard solutions were prepared by dissolving working standards in methanol. Calibration standards were prepared at 8 concentration levels ranging from 0.05 to 500 μg/mL in PBS. 

The cumulative amount of drug that permeated through the skin was plotted as a function of time and the linear portion of the curve was used to calculate the flux and lag time. The permeability coefficient was calculated according to the following equation:(1)$$K_{p}=J_{ss}/C_{0}$$
where *J_ss_* is the steady-state flux of the permeant per unit area (ng/cm^2^/h), *K_p_* is the permeability coefficient (cm/h), and *C*_0_ is the concentration of drug applied to the skin surface (ng/cm^3^).

### 2.7. Skin Permeation Evaluation Using DNDI-0690

In the next stage, the permeation profile of DNDI-0690 from 2 selected FFSs (Eudragit RS PO with DBS and Klucel with TEC) both in and through *Leishmania*-infected and uninfected BALB/c was assessed. The respective placebo FFS (as prepared above) and the control vehicle PG–EtOH (1:1) were spiked with 1% (*w*/*v*) DNDI-0690 and returned to the stirrer plate to allow the drug to mix homogeneously before application. Two circular skin discs (approximately 15 mm in diameter) were obtained per donor mouse; one contained the leishmaniasis nodule and another disc of unaffected skin was collected from the area higher up the back of the mouse. Fat and muscle tissue were carefully removed using forceps, and the skin was gently stretched on Whatman filter paper. The skin was placed between the greased donor and receptor compartment of the Franz cell, followed by similar preparations described in the methodology in 2.6. Next, the 1% DNDI-0690 in FFS or PG–EtOH (1:1) was applied to the skin (130 μL/cm^2^), and 100 μL of receptor solution was replaced with fresh receptor solution at regular time intervals over a period of 54 h and analysed by ultra-high performance liquid chromatography–quadrupole time-of-flight mass spectrometry (UHPLC–qTOF–MS). The cumulative amount, lag time, permeability coefficient, and steady-state flux were calculated as described previously.

### 2.8. DNDI-0690 Skin Drug Distribution 

At the end of the experiment, the Franz cells were dismantled and the donor chambers of the Franz cells were washed with 0.5 mL of acetonitrile–water solution (ACN–H_2_O (1:1). Any drug remaining on the skin surface was removed using a clean dry cotton swab that was subsequently extracted using 1 mL ACN–H_2_O (1:1). The amount of drug in the washing liquid and the cotton swab was quantified using LC–MS/MS. To extract DNDI-0690 from the skin, we transferred the skin disc and 1 mL of ice cold ACN (100%) to a clean tube and left it on a shaking plate (800 rpm) for 1 h before centrifugation at 13,000 rpm for 30 min at 4 °C. A 50 μL aliquot of the supernatant was diluted with an equal volume of water and stored at −70 °C until sample analysis.

### 2.9. DNDI-0690 Quantification by UHPLC-TOF

Sample analysis was performed by UHPLC–qTOF–MS. The instrumentation consisted of an Agilent 1260 infinity II autosampler, Agilent 1290 infinity UHPLC system (1), and an Agilent 6550Q ToF mass spectrometer used in positive ion electrospray mode. Chromatography was performed on a Kinetex 1.7 µm BiPhenyl column (2.1 × 50 mm, Phenomenex, Cheshire, UK). The mobile phase consisted of 0.1% formic acid in water (mobile phase A) and 0.1% formic acid in acetonitrile (mobile phase B). The samples were introduced onto the column using 2% mobile phase B at a flow rate of 0.4 mL per minute, followed by a linear gradient to 95% mobile phase B between 0.3 and 2.9 min. The composition was maintained at 95% mobile phase B until 3.55 min, returning to an initial 2% mobile phase B at 3.6 min. Test compounds eluted between 2.3 and 2.5 min. Acceptable mass balance for total compound recovery was defined as 80 to 110%. 

### 2.10. In Vivo Efficacy Evaluation

When an average nodule size of 3.62 ± 0.80 mm was reached, mice were allocated in groups of 5, ensuring similar average nodule diameters amongst the groups (one-way ANOVA, *p* > 0.05, SPSSv26), and the 10-day treatment was initiated. Each experiment included an untreated control group (*n* = 5), a positive control group (paromomycin, intraperitoneally, 50 mg/kg, QD, *n* = 5), and a DNDI-0690-treated group (50 mg/kg in polyethylene glycol 400, orally, QD, *n* = 5). Topical administration groups included 1% (*w*/*v*) DNDI-0690 in Eudragit-DBS, Klucel-TEC, and PG–EtOH (1:1) (50 µL, BID (twice daily), *n* = 5), as well as the respective vehicle-only controls (50 μL, BID, *n* = 5). The mice were culled a day after the last drug administration, and a circular disc encompassing the nodule was collected. Half of the lesion was weighed, cut into small pieces, and homogenised (Precellys, Bertin Technologies) in 1 mL of PBS using reinforced tubes pre-loaded with stainless steel beads of variable sizes. The DNA of 50 μL of the skin homogenate was extracted using the DNeasy blood and tissue kit (Qiagen) and eluted in a similar volume of extraction buffer; then, 2 μL was used to determine the parasite load by comparing the sample Ct (threshold cycle) with the Ct of previously prepared standards (skin homogenates spiked with 1 × 10^8^
*L. major* and 8 dilutions (1:10)) by quantitative PCR (qPCR). The limit of quantification was established as 10^2^
*L. major* parasites per 50 μL skin homogenate.

## 3. Results

### 3.1. Formulation and Film Evaluation

Upon visual evaluation of the different FFSs, the Kollicoat SR- (a 30% dispersion (aq.) of polyvinvyl acetate stabilised by polyvinyl-pyrrolidone) and Pemulen TR-2 (acrylates/C10–30 alkyl acrylate crosspolymer)-based solutions appeared cloudy, demonstrating only partial solubilisation of the polymer in ethanol, and were eliminated from further investigation. The Kollidon 90F and Eudragit NM30D were excluded from the study as they demonstrated poor film flexibility, as the film presented cracks after arm torsion, and solvent drift, complicating accurate drug dosing, respectively. The visual evaluation of the remaining three polymeric films (Kollidon VA64, Eudragit RS, and Klucel LF) with or without plasticiser (PG, TEC, DBS, and TBC) is summarised in Table 2. Overall, these FFSs received a positive evaluation for viscosity, cosmetic attributes, solvent drift, and outward stickiness. The drying time of the film is important for therapy adherence; all tested films demonstrated drying times inferior to the three-minute cut-off value, except for Klucel with PG, DBS, and TBC that slightly exceeded this time. The film integrity (defined as the ability of the film to resist cracking), which is fundamentally related to the drug–skin contact time and its potential to sustain drug delivery, was demonstrated to be superior for the Eudragit polymeric films. A total of nine polymer–plasticiser combinations (three polymers: Kollidon VA64, Eudragit RS, and Klucel LF, and two plasticisers: TEC and DBS) satisfied the film evaluation parameters and were selected for subsequent hydrocortisone release testing.

### 3.2. In Vitro Hydrocortisone Permeation across the Strat-M Membrane

In order to compare the release patterns from the nine selected FFS films, we selected hydrocortisone (HC), a commonly used steroid to treat dermatological disorders, due to the similarity of its physico-chemical properties to those of DNDI-0690; both are small molecules, have a high partition coefficient (log P _DNDI-0690_ = 2.45, log P _hydrocortisone_ = 1.61), and are poorly water-soluble. 

Figure 1 shows the HC release patterns across the Strat-M membrane. The HC release was highest from the hydrophilic Klucel-based FFSs and ranged from 1162 μg/cm^2^ (Klucel) to 1488 μg/cm^2^ (Klucel-TEC). After a brief lag-time (40 to 60 min; Table 3), HC permeation followed zero-order kinetics (*r^2^* ≥ 0.99 for all Klucel films). The HC permeation was more modest when applied in the hydrophobic Eudragit vehicles; the Eudragit–DBS combination in particular (both hydrophobic) resulted in slow, almost first-order release kinetics after approximately 3 h. The HC release from the Kollidon and Kollidon–DBS films followed a somewhat different trend; an initial HC burst was observed (suggesting no membrane control), followed by a slow HC release. Across the films tested, the permeation fluxes ranked as follows: Eudragit–DBS << Eudragit < Kollidon < Kollidon–TEC< PG–EtOH < Kollidon–DBS < EtOH < Eudragit–TEC << Klucel < Klucel–DBS < Klucel–TEC. Given the similar hydrophobicity of DNDI-0690 and HC, further experiments focused on the FFSs resulting in the most divergent HC release patterns: Eudragit–DBS and Klucel–TEC. 

### 3.3. DNDI-0690 Release and Distribution in Leishmania-Infected and Uninfected Mouse Skin

We evaluated the permeation profile of DNDI-0690 when applied to the skin in hydrophobic (Eudragit–DBS), hydrophilic FFS (Klucel–TEC), and PG–EtOH solvents, in conjunction with the impact of the local *Leishmania* infection on the skin distribution of the drug, by utilising ex vivo diseased and healthy BALB/c mouse skin. Figure 2 shows the cumulative amount of DNDI-0690 that permeated over time and highlights a subtle but statistically significant increase of permeation for infected compared to uninfected skin for both FFSs (repeated-measures ANOVA, *p* < 0.05, SPSS v26, *n* = 4) but not for the PG–EtOH formulation (repeated-measures ANOVA, *p* > 0.05, SPSS v26, *n* = 4). Furthermore, the release from the Eudragit FFS was statistically significantly higher for both infected and uninfected skin compared to the Klucel FFS (Table 4). Again, this difference was not statistically different for the PG–EtOH vehicle.

To investigate whether the different FFSs modified the skin drug distribution, we compared the concentrations of DNDI-0690 extracted from the skin (Figure 3B). The Klucel–TEC formulation resulted in a 4- and 10-fold higher drug amount in infected skin as opposed to the Eudragit FFS and the conventional PG–EtOH formulation, respectively (one-way ANOVA, *p* < 0.05, SPSS v26 (IBM, Hampshire, UK)). In uninfected skin, this difference was even more pronounced and increased to a 45-fold difference compared with the Eudragit FFS and a 14-fold difference for the PG–EtOH vehicle (one-way ANOVA, *p* < 0.05, SPSS v26). Furthermore, the *Leishmania* infection appeared to impact the skin distribution of DNDI-0690 but only when applied in the Klucel film. The amounts of DNDI-0690 extracted from infected versus uninfected skin were not significantly different for the Eudragit FFS or PG–EtOH formulation (one-way ANOVA, *p* > 0.05, SPSSv26).

### 3.4. Efficacy of DNDI-0690 Film-Forming Systems against Experimental Cutaneous Leishmaniasis

Finally, the efficacy of the FFSs was evaluated in vivo in the *L. major* BALB/c mouse experimental model of CL. The lesion size of the mice receiving DNDI-0690 orally or paromomycin intraperitoneally (positive control) was significantly reduced compared to the untreated control (repeated-measures ANOVA, *p* < 0.5, SPSS v26) (Figure 3A). The parasite loads in the skin (measured by qPCR) at the end of the experiment confirmed these observations (Figure 3B). The topical application of DNDI-0690 in FFSs and the conventional formulation (PG–EtOH) (Figure 3C) twice-a-day appeared to halt the lesion size progression. However, this lesion size reduction was not statistically significant when compared to the respective vehicle control groups (repeated-measures ANOVA, *p* > 0.05, SPSS v26). A similar trend was observed when investigating the cutaneous parasite burden for the FFS groups. Only the DNDI-0690–Eudragit group was able to reduce the parasite load compared to the Eudragit vehicle control group. Given the potent in vitro activity of DNDI-0690, these findings suggest that inadequate amounts of drug reach the parasites in the dermal layers of the skin upon topical application. 

Overall, these findings indicate that even though these FFS formulations resulted in higher concentrations of DNDI-0690 in the skin compared to the conventional formulation, the concentrations were insufficient to remove the *Leishmania* parasites from the skin and resolve CL. 

## 4. Discussion

As CL typically occurs on exposed and visible body parts, the first phase of the research focused on selecting cosmetically acceptable FFSs with a short drying time to potentially maximise patient compliance. To further facilitate a touch-free spray-on drug application, we found that low-viscosity vehicles with a controllable drift post-application were likely the most appropriate formulations to ensure the formation of a smooth and thin film upon solvent evaporation, whilst still allowing precise drug dose administration. The flexibility of the films is important for enhancing the skin–film contact that contributes to sustained drug delivery. Film flexibility is enhanced if the glass transition temperature (Tg) of the film polymer is below that of the skin surface [9]. The Tg of Klucel LF is approximately 0 °C [26], which resulted in flexible films. In contrast, a film composed of only Eudragit polymer, which has a Tg of approximately 65 °C [27,28], demonstrated extensive cracking upon skin movement. The addition of plasticisers enhanced film flexibility as was apparent from the lack of visible cracks upon skin stretching. Finally, film removal was investigated as a way to evaluate film integrity. In parallel to the flexibility of the film, the integrity also facilitates skin–drug contact. This parameter is difficult to evaluate, and methodological standards are lacking. Some methodologies evaluate the film integrity visually after overnight “wearing” [29]. Here, the removal of the film was evaluated by wiping it down with lukewarm water in a similar way for the different volunteers and it was possible to identify increased integrity of the Eudragit films in comparison to the other FFSs.

The main aim of the research was to evaluate the use of FFSs to enhance and optimise DNDI-0690 delivery to the dermal layers of the skin to significantly reduce the *Leishmania* parasites in the skin and cure CL. In vitro Franz diffusion experiments demonstrated the lack of permeation of the drug when applied topically as a saturated solution (0.063% (*w*/*v*) DNDI-0690 in Pg–EtOH (1:1))—this is when maximum drug transfer occurs (thermodynamic activity = 1). Even when using dermal microdialysis technology, we were unable to detect any free unbound drug upon a single application of this topical solution to the skin. The advantage of an FFS is that after application to the skin (and as the solvent evaporates), the concentration of the drug in the remaining vehicle increases, potentially creating a transient state of supersaturation either on the skin or in the stratum corneum (thermodynamic activity > 1) [30]. Together with the improved skin–film contact, an FFS could improve dermal drug distribution of a drug. For example, the increased contact time of terbinafine on the skin when applied as FFS gel Lamisil Once permitted sustained delivery of the drug. In fact, seven days after the application, terbinafine was still detectable in the stratum corneum at a level of 23 ng/cm^2^, whereas daily application of a conventional formulation over a period of 7 days resulted in terbinafine levels of 46 ng/cm^2^ [19,31].

In an effort to limit the use of the experimental drug, we designed an experimental strategy to evaluate the permeation profiles of HC, a drug with similar physicochemical properties to DNDI-0690, from the Kollidon, Klucel, and Eudragit films across the Strat-M membrane. This artificial membrane, designed to mimic human skin, provides a cost-effective alternative model to identify permeation trends and rank pharmaceutical formulations, although not to quantify absolute permeability values for human skin. For example, this membrane successfully predicted the effect of multiple penetration-modifying agents on the permeation of nicotine and demonstrated a time point correlation of 90% or more with human cadaver skin [25]. In our study, HC demonstrated a higher permeation flux when applied in the hydrophilic Klucel film compared to a slower release from the hydrophobic Eudragit films. Similar findings were obtained when investigating the impact of FFSs on the release profile of ethinyl estradiol through heat-separated human epidermis [29] or betamethasone 17-valerate across a silicone membrane. In the latter example, Klucel films released betamethasone 17-valerate with consistently high zero-order kinetics, whereas a slow release with an initial burst was observed when using the Eudragit FFS [9].

An opposite trend was observed when we evaluated the percutaneous permeation profile of DNDI-0690 across BALB/c mouse skin; the permeated cumulative amount across both infected and healthy skin was higher for the Eudragit than the Klucel film. Interestingly, the Klucel film was superior in depositing more DNDI-0690 in the skin when compared to the Eudragit film or the Pg–EtOH vehicle, which is potentially more important as this is where the parasites reside. The reservoir effect of the skin is well recognised [30]—lipophilic drugs demonstrate a high affinity for the hydrophobic stratum corneum and are quickly released from the hydrophilic Klucel–TEC film. For the DNDI-0690 Klucel FFS, we also observed higher drug amounts in uninfected skin compared to infected skin in contrast to the higher concentrations extracted from the infected compared to healthy skin for the DNDI-0690 Eudragit FFS. This observation highlights the interaction between the physico-chemical properties of the drug, the nature of the polymer, and the microenvironment of the parasite [32]. The hydrophobic DNDI-0690 permeates faster across the infected skin, which has been shown to be more permeable to both hydrophilic and hydrophobic drugs [33,34] but forms a smaller reservoir in infected skin, potentially due to the inflammation-associated oedema in the skin. The disadvantage of water-soluble films such as Klucel is that they tend to be less resistant to removal by washing or perspiration (as observed in our film integrity results), leaving little time to establish the in-skin reservoir [35]. This might explain why the DNDI-0690 Eudragit film with its high integrity and steady permeation rate was the only FFS that resulted in a significant reduction of the parasite load when applied onto the CL nodule in experimental CL.

A limitation of this work is the use of mouse skin, which has significant differences to human skin in terms of thickness and number of hair follicles and generally results in a higher permeability. Nevertheless, the impact of the different FFSs on the release of DNDI-0690 and its skin distribution were compared using skin from infected nodules and uninfected mouse skin in line with previous in vivo studies, aiming to reflect a trend rather than absolute differences. Due to the irritant nature of the solvents used, this work also reflected product usage on early CL nodules rather than on open ulcers. The latter is likely to require the usage of less irritant solvents.

Overall, this is the first report that describes the use of FFS to modify the topical delivery of an antileishmanial drug to treat CL. Whereas only one FFS was able to reduce the parasite load in the skin, this study was able to demonstrate that the nature of the polymer and film not only influenced the cosmetic attributes of the film, but it was also able to modulate drug release, targeting different skin drug distribution profiles depending on the skin microenvironment. In conclusion, careful modulation of formulation could aid the topical delivery of DNDI-0690; topical formulation efforts should focus on other promising drug candidates in order to deliver an affordable, topical formulation to strengthen the treatment arsenal for CL.

## Figures and Tables

**Figure 1 pharmaceutics-13-00516-f001:**
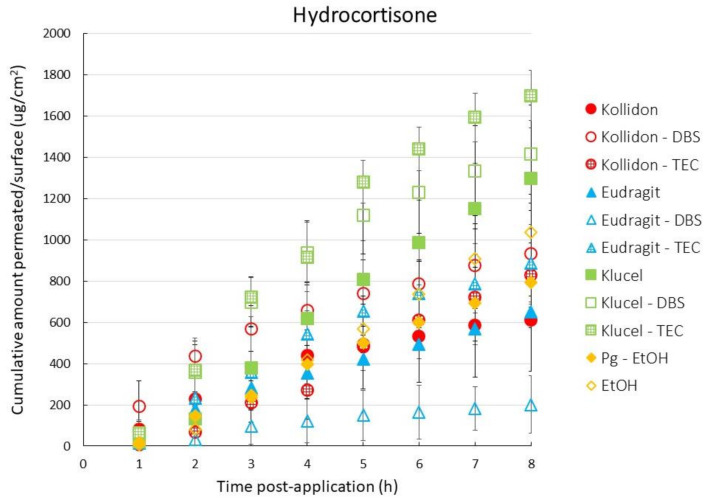
Cumulative amount of hydrocortisone (HC) that permeated across the Strat-M membrane per surface area (ug/cm^2^) in time (*n* = 4, average ± SD) upon application using different formulations (triethyl citrate (TEC), tributyl citrate (TBC), dibutyl sebacate (DBS), propylene glycol (PG) and ethanol (EtOH)).

**Figure 2 pharmaceutics-13-00516-f002:**
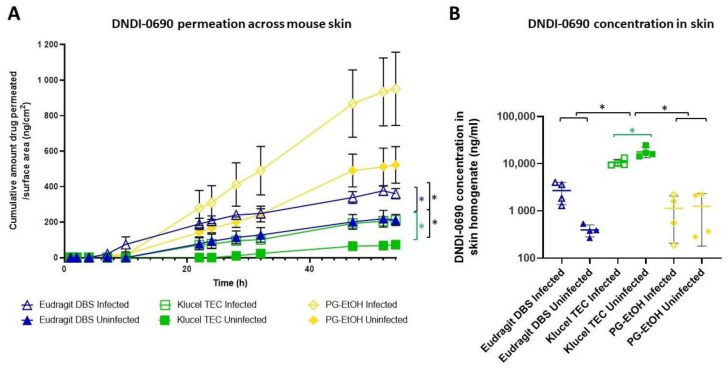
(**A**) Cumulative amount of DNDI-0690 permeated per surface are in time (*n* = 4, average ± SEM), green and blue bracket for uninfected vs. infected skin for Klucel and Eudragit, respectively; black bracket Eudragit vs. Klucel for infected skin and Eudragit vs. Klucel for uninfected skin (* *p* < 0.05, repeated-measures ANOVA, post-hoc Tukey test, SPSS v26). (**B**) DNDI-0690 concentration present in 50 μL skin homogenate (*n* = 4, average ± SD) (* *p* < 0.05, one-way ANOVA, post-hoc Tukey test, SPSS v26).

**Figure 3 pharmaceutics-13-00516-f003:**
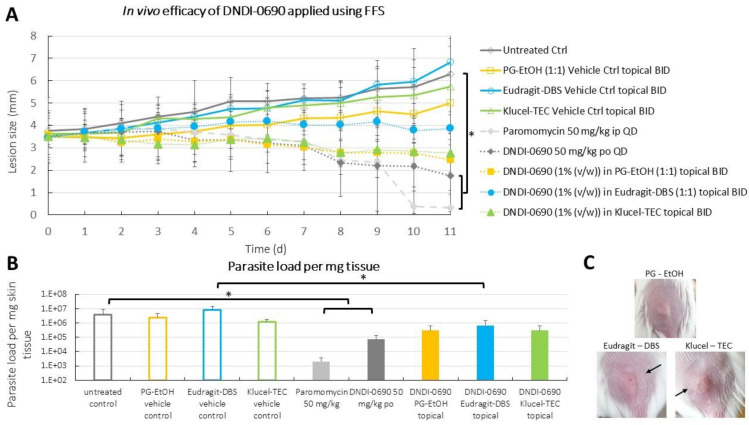
The in vivo efficacy of DNDI-0690 FFS formulations in an experimental model of cutaneous leishmaniasis (CL). (**A**) The skin lesion sizes of *Leishmania*-infected mice were measured daily throughout the treatment that started on day 1 and terminated on day 11. This graph shows the lesion size (mm) in function of time (*n* = 4, average ± SD) (* *p* < 0.05, repeated-measures ANOVA, post-hoc Tukey test, SPSSv26) (po, per os; ip, intraperitoneally; BID, twice daily; QD, once daily). (**B**) The average parasite load (analysed by qPCR) in the CL lesions per treatment group (*n* = 4, average ± SD) (* *p* < 0.05, one-way ANOVA, post-hoc Tukey test, SPSS v26). (**C**) The DNDI-0690 containing Eudragit–DBS and Klucel–TEC film (bottom panel, black arrow indicates the edge of each film) and the DNDI-0690 in PG–EtOH suspension (top) upon application on the *Leishmania*-infected BALB/c mouse skin.

**Table 1 pharmaceutics-13-00516-t001:** Composition of the placebo film-forming system (FFS).

Film-Forming System	Polymer		Plasticiser	EtOH		H_2_O	
	% *w*/*v*	Weight (g)	Weight (mg)	Weight (g)	%	Weight (g)	%
Kollidon 90F	7.5	0.375		4.625	92.5		
With plasticiser (PG/DBS/TBC/TEC)	7.5	0.375	75	4.55	91		
Kollidon VA64	10	0.5		4.5	90		
With plasticiser (PG/DBS/TBC/TEC)	10	0.5	100	4.4	88		
Kollicoat SR 30D	15	0.75		4.25	85		
With plasticiser (PG/DBS/TBC/TEC)	15	0.75	150	4.1	82		
Pemulen TR-2 NF	2	0.1		4.9	98		
With plasticiser (PG/DBS/TBC/TEC)	2	0.1	20	4.88	97.6		
Eudragit RS PO	10	0.5		4.25	85	0.250	5
With plasticiser (PG/DBS/TBC/TEC)	10	0.5	100	4.15	83	0.250	5
Eudragit NM30 D	7.5	0.375		4.125	82.5	0.5	10
With plasticiser (PG/DBS/TBC/TEC)	7.5	0.375	75	4.05	81	0.5	10
Klucel LF	5	0.25		4.75	95		
With plasticiser (PG/DBS/TBC/TEC)	5	0.25	50	4.7	94		

PG = propylene glycol (log P = −0.47), dibutyl sebacate (log P = 4.87), tributyl citrate (log P = 2.83), triethyl citrate (log P = 0.12). Log P values calculated using ChemDraw Prime software v20.0.

**Table 2 pharmaceutics-13-00516-t002:** Visual evaluation of film-forming systems containing different types of plasticiser.

Polymers and Plasticiser		Appearance	Viscosity	Drying Time	Stickiness	Cosmetic Attributes		Film Flexibility	Film Removal	
		Clear/Opaque	Water (1) Glycerol (2) Lamisil (3)	(Min)	+ (Low) ++ (Medium) +++ (High)	Clear/Opaque	Vehicle Drift - (Absent) + (Present)	Cracking - (Absent) + (Low) ++ (Medium) +++ (High)	With Water + (Easy) ++ (Medium) +++ (Difficult)	Peeling - (Absent) + (Present)
10% Kollidon VA64	None	clear	1	<2 min	+	clear	-	+	+	-
	PG	clear	1	<2 min	+	clear	-	-	+	-
	DBS	clear	1	<2 min	+	clear	-	+	+	-
	TBC	clear	1	<2 min	+	clear	-	-	+	-
	TEC	clear	1	<2 min	+	clear	-	-	+	-
10% Eudragit RS PO	None	clear	1	<2 min	+	clear	+	++	+++	-
	PG	clear	1	<2 min	+	clear	+	+	+++	-
	DBS	clear	1	<2 min	+	clear	+	-	+++	-
	TBC	clear	1	<2 min	+	clear	-	-	+++	-
	TEC	clear	1	<2 min	+	clear	-	-	++	+
5% Klucel LF Pharm	None	clear	1–2	2–3 min	+	clear	-	-	+	-
	PG	clear	1–2	3–4 min	+	clear	-	-	+	-
	DBS	clear	1–2	3–4 min	+	clear	-	-	+	-
	TBC	clear	1–2	3–4 min	+	clear	-	-	+	-
	TEC	clear	1–2	2–3 min	+	clear	-	-	+	-

**Table 3 pharmaceutics-13-00516-t003:** Permeation parameters for hydrocortisone (HC) when applied to the Strat-M membrane (average ± SD, *n* = 4).

Formulation		Flux (×10^3^ ng/cm^2^/h)	Lag Time (h)	Kp (cm/h)
Eudragit	Alone	88 ± 36	0.08	8.81 × 10^−6^
	DBS	33 ± 26	0.84	3.35 × 10^−6^
	TEC	131±36	0.07	16.0 × 10^−6^
Klucel	Alone	196 ± 54	1.03	19.6 × 10^−6^
	DBS	254 ± 35	0.63	25.4 × 10^−6^
	TEC	279 ± 92	0.67	27.9 × 10^−6^
Kollidon	Alone	92 ± 14	0	9.15 × 10^−6^
	DBS	117 ± 67	0	11.7 × 10^−6^
	TEC	113 ± 18	1.19	11.3 × 10^−6^
PG–EtOH		114 ± 60	0.69	11.4 × 10^−6^
EtOH		140 ± 30	1.17	14.0 × 10^−6^

**Table 4 pharmaceutics-13-00516-t004:** Permeation parameters for DNDI-0690 when applied to *Leishmania*-infected or healthy BALB/c mouse skin (average ± SD, *n* = 4).

Formulation	Skin Status	Cumulative Amount/Surface (ng/cm^2^)	Flux (ng/cm^2^/h)	Lag Time (h)	Kp (cm/h)
Eudragit–DBS	Infected	360 ± 58	7.83 ± 1.22 *	0.14	78.3 × 10^−6^
	Uninfected	212 ± 62	5.07 ± 1.44 *	7.91	50.7 × 10^−6^
Klucel–TEC	Infected	212 ± 47	4.86 ± 0.92 **	8.47	48.6 × 10^−6^
	Uninfected	72 ± 18	2.44 ± 0.37 **	23.20	24.4 × 10^−6^
PG–EtOH	Infected	642 ± 289	14.48 ± 6.81	9.78	14.5 × 10^−6^
	Uninfected	522 ± 206	11.83 ± 4.73	8.89	11.8 × 10^−6^

* *p* < 0.05, Eudragit–DBS infected vs. uninfected; ** *p* < 0.05, Klucel–TEC infected vs. uninfected.

## Data Availability

The data presented in this study are available on request from the corresponding author.

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
