# Peer review of "Film-Forming Systems for the Delivery of DNDI-0690 to Treat Cutaneous Leishmaniasis"

_pharmaceutics, 2021, doi:10.3390/pharmaceutics13040516_

Round 1

Reviewer 1 Report

Reviewer’s Comments to Author 

The authors have carried out an investigation on film-forming systems for cutaneous Leishmaniasis. Overall, the manuscript submitted by Katrien et al. is of great interest to me. The article is comprehensive, well-organized, and well referenced. The reviewer thinks that a potential reader can easily understand the details by consulting the main data outlined in the manuscript. The abstract accurately conveys the whole content of the article and the conclusions are fully appropriate in reviewer’s understanding. The writing is clear and to the point, which is, crucial for a research article. Regarding the writing style is understandable and easy to read for the readership of journal. Tables are also clear and provide a full overview of the trials discussed in the text. For this reason, I consider current version of manuscript adequate to be published in Pharmaceutics.  

Minor modification

Title of the manuscript is very general and can point to a misleading horizon. This should be rephrased and should include a wording such as: Film-forming systems for cutaneous Leishmaniasis for the delivery of DNDI-0690.

Author Response

We thank the reviewer for consideration of the manuscript and support for publication. 

The title was amended to: Film-Forming Systems for the delivery of DNDI-0690 to treat Cutaneous Leishmaniasis.

Reviewer 2 Report

The manuscript deals with a new formulative attempt of cutaneous administration of the poorly water soluble DNDI-0690 to treat single and closed cutaneous leishmaniasis nodules in BALB/c mice. Drug saturated solution was already tested in Franz diffusion studies utilizing ex vivo mouse skin, but the lack of DNDI-0690 permeation through both Leishmania-infected and uninfected skin was reported. The authors explored the use of film forming system (FFS) as a skin delivery system. Unfortunately, they had no success as the findings indicate that even though these FFS formulations resulted in higher concentrations of DNDI-0690 in the skin compared to the conventional formulation, the concentrations were insufficient to remove the Leishmania parasites from the skin and resolve CL.

In addition to logD, which is not unfavorable for skin permeation, the other available physico-chemical characteristics of DNDI-0690 should be added.

The characterization of the FFS is quite weak, mainly based on visual observation. Since well characterized systems, in terms of viscosity, drying time, texture analysis, thermogravimetric evaluation, are reported in literature, further investigations should be added.

“Materials and methods”

  • Paragraph 2.4: it should be moved after 2.5
  • Paragraph 2.5: how was viscosity evaluation performed? Solvent evaporation should be measured.
  • Paragraph 2.6: why are samples stored at -70° before analysis? Considering the prepared calibration curve ranging from 0.05 to 500 μg/ml, is it accurate enough?
  • Paragraph 2.7: line 228-231 can be omitted as already described before; line 235-240 should be considered Data analysis useful also for the use of the Strat M. Organize paragraphs 2.6 and 2.7 better for the common parts.

In 3.1 it should be clarified what the authors mean with the term “film integrity”; this description is not reported above.

The use of Hydrocortisone for the preliminary evaluation is questionable. Control solution containing DNDI-0690 should be investigated.

Table 2 is not easy to read; too much information is included.

In Discussion, line 440-446, comments are not clear; add what BMV is.

Table 3 and 4 are not cited in the text.

Moderate English changes required; check some typos.

Author Response

We thank the reviewer for corrections and comments which we have addressed and now provide an improved revised manuscript.  Please find our response to your comments below.

In addition to logD, which is not unfavorable for skin permeation, the other available physico-chemical characteristics of DNDI-0690 should be added.

Response: These have now been included in the manuscript.

The characterization of the FFS is quite weak, mainly based on visual observation. Since well characterized systems, in terms of viscosity, drying time, texture analysis, thermogravimetric evaluation, are reported in literature, further investigations should be added.

Response: The proof-of-concept results that were published in this manuscript were obtained following similar analyses and methodologies to those already published in previous peer-reviewed manuscripts (Frederiksen et al, 2016). Some of the properties such as viscosity are difficult to measure as solvent evaporation will change the properties of the formulation. In line with the reviewer’s suggestion, a more thorough characterisation is planned as future work.

“Materials and methods”

Paragraph 2.4: it should be moved after 2.5.

Response: This section was moved.

Paragraph 2.5: how was viscosity evaluation performed? Solvent evaporation should be measured.

Response: This section has been amended to: “The viscosity, an important factor for spray formulations affecting dosing accuracy and ability of the formulation to spread on the skin, was compared visibly to the viscosity of water (low – score 1), glycerol (medium – score 2) and the benchmark FFS Lamisil® Once (high – score 3)”.

Paragraph 2.6: why are samples stored at -70° before analysis?

Response: This section was now corrected.

Considering the prepared calibration curve ranging from 0.05 to 500 μg/ml, is it accurate enough? Response: This HPLC method was validated; R2 for this standard curve was above 0.996 for all runs (including the runs used to determine intra and inter-variability). The concentrations of the samples ranged from 75 ng to 250 ug/ml and all but 12 samples (out off 440) fell below the limit of quantification at time point 1 and were as such returned in the calculation as not quantifiable.

Paragraph 2.7: line 228-231 can be omitted as already described before; line 235-240 should be considered Data analysis useful also for the use of the Strat M. Organize paragraphs 2.6 and 2.7 better for the common parts.

Response: These sections were shortened and simplified.

In 3.1 it should be clarified what the authors mean with the term “film integrity”; this description is not reported above.

Response: This has been clarified as requested.

The use of Hydrocortisone for the preliminary evaluation is questionable. Control solution containing DNDI-0690 should be investigated.

Response: The influence of each film-forming system on the permeation performance of the skin model permeant hydrocortisone across the Strat-M membrane was used to rank order the different selected film-forming systems. Due to the limited availability of DNDI-0690 at the start of the project, hydrocortisone was used given its physico-chemical similarities to DNDI-0690. The Strat-M membrane was used in an effort to limit the use of the animal tissue. Previous permeation reports discussed the limitations of using such membranes and warns that findings might not translate to human or animal skin. We therefore opted to investigate the DNDI-0690 permeation across the actual infected and uninfected skin membrane that relates to the in vivo investigation rather than the Strat-M membrane. We recognise that a study of the fundamental permeation behaviour of DNDI-0690 using other membranes would provide important information but this was not the main aim of this research.

Table 2 is not easy to read; too much information is included.

Response: This table has been re-organised.

In Discussion, line 440-446, comments are not clear; add what BMV is.

Response: his has now been amended.

Table 3 and 4 are not cited in the text.

Response: Appropriate references to these tables were now included in the text.

Moderate English changes required; check some typos.

Response: The text was re-read and amended; typos were corrected.

Round 2

Reviewer 2 Report

Manuscript accepted in present form

Author Response

We thank the reviewer for consideration of the manuscript and support for publication.